# A Digital Cash Paradigm with Valued and No-Valued e-Coins

**Ricard Borges** [1,2,†] and **Francesc Sebé** [1,2,*,†]

---

1   Department of Mathematics, Universitat de Lleida, E-25001 Lleida, Spain; rborges@matematica.udl.cat
2   Cybercat: Center for Cybersecurity Research of Catalonia, E-25001 Lleida, Spain
*   Correspondence: francesc.sebe@udl.cat; Tel.: +34-973-702-713
†   These authors contributed equally to this work.

**Abstract:** Digital cash is a form of money that is stored digitally. Its main advantage when compared to traditional credit or debit cards is the possibility of carrying out anonymous transactions. Diverse digital cash paradigms have been proposed during the last decades, providing different approaches to avoid the double-spending fraud, or features like divisibility or transferability. This paper presents a new digital cash paradigm that includes the so-called no-valued e-coins, which are e-coins that can be generated free of charge by customers. A vendor receiving a payment cannot distinguish whether the received e-coin is valued or not, but the customer will receive the requested digital item only in the former case. A straightforward application of bogus transactions involving no-valued e-coins is the masking of consumption patterns. This new paradigm has also proven its validity in the scope of privacy-preserving pay-by-phone parking systems, and we believe it can become a very versatile building block in the design of privacy-preserving protocols in other areas of research. This paper provides a formal description of the new paradigm, including the features required for each of its components together with a formal analysis of its security.

**Keywords:** cryptography; digital cash; privacy





## 1. Introduction

The European Commission defines digital cash (also referred to as *e-money* or *e-cash*) as a digital alternative to cash. It allows users to make cashless payments with money stored on a card or a phone, or over the Internet.

Digital cash was first proposed in the early 1980s by Chaum [1] in a proposal based on the newly invented blind signature cryptographic primitive. That proposal includes three actors: the *bank*, the *payer* (the *customer*), and the *payee* (the *vendor*), and three protocols: *withdraw*, *spend*, and *deposit*. An e-coin is a random sequence of data which has been digitally signed by the bank. During the withdraw protocol, the payer generates a random sequence and, after paying for it, asks the bank to digitally sign it through a blind signature protocol. This ensures that the bank does not learn any information about the issued e-coin so that the payer will be able to spend it anonymously in the future. An e-coin is spent by means of a process by which the payer transmits it to the payee. Finally, when the payee deposits an e-coin, they receive its monetary value from the bank.

Chaum's proposal [1] is both *anonymous* (the identity of the payer is not revealed during payment) and *unlinkable* (it is not possible to determine whether two payments were made by the same payer or not).

A transaction using a digital currency occurs entirely digitally. This means that a dishonest payer could store a copy of an e-coin after having spent it and then try to spend it again in the future. This is the *double-spending* fraud. Two main strategies have been proposed to cope with attempts to double-spend. In [1], upon receiving a payment, the payee asks the bank to check that the received e-coin has not been spent before. This is an *on-line* double-spending fraud prevention strategy. In the *off-line* alternative [2], double spending is not checked during the payment process, but it can be detected later when the

e-coin is deposited. In cases of double-spending, the anonymity of the double-spender is lifted so that they can be prosecuted.

Research on digital cash systems has addressed diverse features such as e-coin *divisibility,* which allows a user to withdraw a coin and spend it several times by dividing its value (up to a given limit). The first practical divisible digital cash system was proposed in [3]. That proposal is anonymous but it is possible to link several spends from a single divisible coin. The system in [4] is divisible and unlinkable but the vendor and the bank know which part of the coin is being spent. Hence, unlinkability is provided partially. The first divisible and strong unlinkable e-cash system was described in [5]. A more efficient system in terms of computational and communications cost was proposed later in [6]. Research in the divisibility feature is still active with proposals such as [7], in which a large e-coin can be divided into several small ones with arbitrarily integer values. The system is efficient as both the pay and deposit procedures run in constant time. The authors of [8] provide a formal and complete security model for divisible e-cash and study constructions based on pseudo-random functions.

Another research line focused on digital cash systems able to deal with low value payments, that is, *micropayments* [9]. These systems have been addressed by the research community with proposals aiming to reduce the number of public key operations by replacing them with lightweight hash computations. The Micromint system [9] can be seen as a precursor of the widely-known *proof-of-work* concept in modern cryptocurrencies. Although digital cash systems need to be efficient both in terms of computation and communication costs, the ever increasing capacity of computers and mobile devices has eliminated the need for so many lightweight proposals. For instance, the authors of [10] achieve a reduced cost by using elliptic curve public key cryptography.

*Transferability* is the most difficult feature of paper cash to be achieved in the digital world. Transferable digital cash systems enable the payee of a transaction to spend the received e-coin in another transaction in which they will play the payer role. In such systems, preventing the double-spending fraud is a hard issue since several people are in possession of the same e-coin throughout its lifetime. The authors of [11] propose a system in which transactions involve the participation of a verifying authority, which checks that e-coins have not been spent before. A recent paper [12] analyzes previous models for transferable digital coins and concludes that they are incomplete. Then, the authors propose a new model and prove its feasibility by giving a concrete construction and rigorous proofs that it satisfies the model.

Modern approaches to digital cash aim to avoid the need for a trusted central entity. This is achieved through the existence of a distributed public ledger storing a record of all the transactions. This is the case for the Bitcoin [13] or the Ethereum [14] cryptocurrencies among many others. Distributed ledgers require the existence of a method for validating transactions without the need to trust a central authority, namely a *consensus mechanism*. The *proof-of-work* approach implemented by Bitcoin has been criticized due to the huge electricity consumption it involves. The *proof-of-stake* alternative, adopted by Ethereum, is much more energy efficient. The design and evaluation of alternative consensus mechanisms is an active area of research.

Attempts to develop a full cash-like digital payment system which is both anonymous, off-line, and secure against double-spending have been forced to include some trusted hardware element. This is the case for the OPERA system [15], which requires a new concept of memory called ORM (one-time-readable memory). The European Commission has reached a similar conclusion to that mentioned in its 'Report on digital euro' [16], which mentions the deployment of two systems in parallel: one based on trusted hardware that can be off-line, anonymous, and without third-party intervention; and an account-based on-line, which is fully software based but excludes the possibility of anonymity.

This paper provides a formal description of a new digital cash paradigm, which enables customers (the payers) to issue no-valued e-coins. These no-valued e-coins are indistinguishable from valued ones and can be used to conduct bogus transactions against

the vendor (the payee). The vendor cannot determine whether the received e-coin was valued or not while the customer receives the requested digital item only when a valued e-coin has been spent.

This paradigm has proven its validity in the scope of pay-by-phone parking applications. In [17], a driver, after parking their car in a regulated zone, acquires tickets for some consecutive short time intervals during which their car is expected to be parked. So as to mask the expected parking duration, all the drivers always request the same amount of tickets. Those tickets belonging to intervals after the expected parking duration are paid through no-valued e-coins. An ad-hoc construction belonging to this digital cash paradigm was presented in [17]. Nevertheless, that initial design assumed the use of a fixed suite of cryptosystems due to restrictions in what concerns to the key-size of the used cryptography. More precisely, the cleartexts of the cryptosystem used by the vendor for signing the issued e-coins had to be large enough to accommodate public keys of the cryptosystem used for encrypting the acquired digital item. The role of the mentioned cryptosystems will be explained later in Section 4.

In this paper, we propose how the mentioned key size limitation can be eliminated. The novel construction also avoids the need in [17] for a timestamp authority. Instead, customers timestamp their transactions by themselves.

Section 1 has presented an introduction to digital cash systems. Section 2 briefly reviews the cryptographic tools required in our construction. Next, Section 3 presents a construction based on the use of OAEP (Optimal Asymmetric Encryption Padding [18]), which allows the simulation of digital signatures over messages of arbitrary length. After that, the novel digital cash paradigm is detailed in Section 4. Some cryptosystems providing the required features of the new paradigm are discussed in Section 5. Section 6 is devoted to analyzing the security of the proposal. Next, experimental results are summarized in Section 7, while Section 8 concludes the paper.

## 2. Preliminaries

This section provides a brief introduction to the cryptographic primitives used by the proposed paradigm.

### 2.1. Public Key Encryption

In a public key cryptosystem [19], each party is in possession of a private-public key pair. The private key is kept secret while the public one can be made worldwide available. A message encrypted under the public key can only be decrypted by a party in possession of the private one.

### 2.2. Digital Signatures

Public key cryptography enables the computation of digital signatures [19]. Given $M$ (usually the hash digest of the data to be signed), the signer computes a digital signature over $M$ using their private key. The resulting signature is denoted as $\text{Sign}_{PK}(M)$, and $\{M, \text{Sign}_{PK}(M)\}$ is a digest-signature tuple.

Such a tuple is validated under signer's public key, $PK$. A positive validation provides integrity, authentication and non-repudiation to the data hashed into $M$.

### 2.3. Simulatable Digital Signatures

The term *simulatable* is widely used in the realm of zero-knowledge proofs [20]. A simulator is an entity which, without knowing the secret, can produce a transcript that looks like a proper interaction between a honest prover and the verifier.

In the context of digital signatures, we say a signature scheme is simulatable, when an entity not knowing the secret key is able to produce valid digest-signature tuples. The digest component of a simulated tuple cannot be chosen by the simulator, otherwise such signature scheme would be forgeable. Use of a simulated tuple for authentication purposes

further requires finding a piece of data whose digest matches the obtained one. This is unfeasible if an appropriate one-way hash function is being used for digest computation.

### 2.4. Blind Signatures

Blind signatures [1] are computed through a protocol run between two parties: Alice, who is in possession of a piece of data whose hash digest is $M$, and Bob who owns a key-pair.

After running the protocol, Alice gets Bob's signature on $M$, while Bob does not learn any information about $M$ nor about the resulting signature.

## 3. Message Digests for Simulatable Signatures

As it will be explained next, our digital cash paradigm requires a signature system allowing the computation of simulated digest-signature tuples, which can be linked to a piece of data. So as to make it possible, the one-way hash function employed in traditional digital signature schemes must be replaced with a similar function allowing some degree of pseudo-random reversibility. In this section we propose a construction fulfilling this requirement.

### 3.1. Optimal Asymmetric Encryption Padding

Optimal Asymmetric Encryption Padding (OAEP) is a procedure initially proposed to pad the plaintext prior to its asymmetric encryption [18]. We next describe OAEP when no plaintext-awareness is required (by setting parameter $k_1$ of the original proposal to 0). Let $m$ be the bitlength of the input plaintext, and let $k_0$ be an integer parameter. OAEP is constructed from two of oracles,

$\mathcal{G}$ and $\mathcal{H}$, producing $m$ and $k_0$ bit outputs, respectively.

The padding procedure takes as input the original plaintext $M$ and a random $k_0$-bit string $r$:

1. Compute $X = M \oplus \mathcal{G}(r)$,
2. Compute $Y = r \oplus \mathcal{H}(X)$,
3. Return $(X, Y)$.

We will denote this process as $(X, Y) = \text{OAEP}_{m,k_0}(M, r)$. The resulting $X$ and $Y$ are $m$ and $k_0$ bits long, respectively. Along the paper, sub-indices $m$ and $k_0$ will be removed when deemed redundant. The reverse procedure returns the original $(M, r)$ pair from $(X, Y)$:

1. Compute $r = Y \oplus \mathcal{H}(X)$,
2. Compute $M = X \oplus \mathcal{G}(r)$,
3. Return $(M, r)$.

We will denote the reverse process as $(M, r) = \text{OAEP}^{-1}_{m,k_0}(X, Y)$.

The following lemma states a property of OAEP, which is crucial for our construction.

**Lemma 1.** *Given a message $M$ and a $k_0$-bit string $Y$, it is hard to find an $\{r, X\}$ pair so that $(X, Y) = \text{OAEP}_{m,k_0}(M, r)$.*

**Proof.** Given $M$ and $Y$, one must find a bitstring $r$ satisfying Equation (1).

$$Y = r \oplus \mathcal{H}(M \oplus \mathcal{G}(r)). \tag{1}$$

Since $\mathcal{G}$ acts as a random oracle, after choosing $r$, the output of $\mathcal{G}(r)$ is assumed to be random so that $M \oplus \mathcal{G}(r)$ is also random. Function $\mathcal{H}$ is also a random oracle so that $\mathcal{H}(M \oplus \mathcal{G}(r))$ is random and so $r \oplus \mathcal{H}(M \oplus \mathcal{G}(r))$ is. Hence, the probability that the resulting random string matches $Y$ is $2^{-k_0}$ so that the expected number of trials needed for finding such an $r$ is $2^{k_0}$ which is unfeasible if $k_0$ is large enough. Given that $r$ is $k_0$ bits long, there are exactly $2^{k_0}$ candidates so that such an $r$ may not exist.

Another possibility is to search for an $X$ satisfying Equation (2).

$$M = X \oplus \mathcal{G}(\mathcal{H}(X) \oplus Y). \tag{2}$$

In this case, an equivalent analysis can be applied leading to a $2^{-m}$ success probability, which is harder, as typically $m \gg k_0$. $\square$

### 3.2. Plaintext Awareness

Given a random $(X, Y)$ pair, the result of computing $(M, r) = \text{OAEP}^{-1}_{m,k_0}(X, Y)$ produces a pseudo-random output. A party performing this computation cannot determine whether $(X, Y)$ was generated at random or it was obtained by computing $(X, Y) = \text{OAEP}_{m,k_0}(M, r)$ from an input $(M, r)$ pair. In this latter case, the creator of $(X, Y)$ was aware of plaintext $M$.

Plaintext awareness is provided by appending $k_1$ '0' bits to $M$ before running the OAEP process [18]. After running the OAEP reverse procedure, one must check that the obtained message $M$ carries $k_1$ attached '0' bits which can then be removed. This construction provides plaintext awareness with probability $1 - 2^{-k_1}$ so that a large enough value for $k_1$ leads to a close to 1 probability.

Throughout this paper, subscript PA will be used to denote that OAEP is being used with the plaintext awareness feature. When plaintext-awareness is provided, string $X$ is $k_1$ bits longer than $M$.

### 3.3. Proposed Construction

We next propose an OAEP-based construction allowing the simulation of digest-signature tuples in such a way that the digest component of simulated tuples can be linked to a piece of data whose length can be chosen by the simulator, but its actual value cannot.

Given an existing simulatable signature scheme, the construction is as follows:

1. Let $M$ be the $m$-bit message to be signed.
2. Let $l$ be the length of the digests signed by the signature scheme.
3. Generate a random $l$-bit bitstring $r$ and compute $(X, Y) = \text{OAEP}_{m,l}(M, r)$.
4. Compute a digital signature over $Y$, namely $\text{Sign}(Y)$.
5. Send the $\{X, Y, \text{Sign}(Y)\}$ tuple to the receiver.

Such a tuple is validated as follows:

1. Validate the $\{Y, \text{Sign}(Y)\}$ digest-signature tuple under signer's public key.
2. Compute $(M, r) = \text{OAEP}^{-1}_{m,l}(X, Y)$ so as to get message $M$.

Note that a tuple $\{X, Y, \text{Sign}(Y)\}$ can be transformed into $\{X', Y, \text{Sign}(Y)\}$ with $X' \neq X$ while the signature validation still produces a positive result. In such a case, the computation of $(M', r') = \text{OAEP}^{-1}_{m,l}(X', Y)$ leads to a piece of data $M' \neq M$. This is not an issue in our construction as long as $M'$ cannot be chosen by the manipulating party, as it has been stated in Lemma 1. The length of $M'$ corresponds to that of $X'$ (or it is $k_1$ bits shorter if plaintext-awareness has been set).

Similarly, any party can generate a simulated $\{Y', \text{Sign}(Y')\}$ tuple for the underlying signature scheme and then choose any $X'$ component. The resulting $\{X', Y', \text{Sign}(Y')\}$ tuple will also result in a positive validation. As in the previous remark, this is not an issue since the resulting $M'$ is obtained pseudo-randomly.

The simulatability of the proposal can be disabled by setting the plaintext awareness feature to OAEP. This is because the pseudo-random piece of data $M'$ obtained by simulation does not meet this feature.

## 4. Novel Digital Cash Paradigm Description

This section provides a detailed description of the proposed digital cash paradigm.

*4.1. Overview*

Our proposal is a pre-paid digital cash paradigm. Customers acquire valued e-coins in advance and store them in an e-wallet. They will later be spent against the vendor when purchasing items. The paradigm is composed of two actors:

- *Vendor.* A vendor sells digital products online and participates in the issuance of valued e-coins after being paid for them.
- *Customers.* They manage an e-wallet containing valued e-coins. These e-coins are acquired in advance and stored until spent during a purchase procedure. Customers can generate no-valued e-coins on their own.

No-valued e-coins can be spent against the vendor but, in such a case, the customer will not receive any product back. No-valued e-coins enable bogus purchases aiming to mask consumption patterns. The vendor cannot distinguish whether the e-coin involved in a transaction was valued or not.

*4.2. e-Coin Composition*

Given vendor's public key, $PK_V$, an e-coin is a tuple of the form represented in Equation (3). Components subindexed with $S$ refer to a cryptographic key-pair used for 'Signing' a transaction. The mentioned key-pair is used by the customer to issue a signature when the e-coin is spent. Those tuple components subindexed with $R$ are related to a key-pair used for 'Receiving' the acquired digital item. Digital items are encrypted by the vendor under public key $Q_R$ before transmitting them to customers.

$$\{v_S, Q_S, (X_S, Y_S), v_R, Q_R, (X_R, Y_R), Y, \text{Sign}_{PK_V}(Y).\} \tag{3}$$

All the e-coin components but the last one ($\text{Sign}_{PK_V}(Y)$) are always generated by the customer. If the e-coin is valued, signature $\text{Sign}_{PK_V}(Y)$ is computed by the vendor; otherwise, it is simulated by the customer. Regarding the components of an e-coin,

- $v_S/Q_S$ is a private/public key-pair of a public key cryptosystem allowing digital signature computation. Hence, data signed with $v_s$ can be validated under $Q_S$. $Q_S$ has been $\text{OAEP}_{PA}$-encoded (with plaintext-awareness) into $(X_S, Y_S) = \text{OAEP}_{PA}(Q_S, r_S)$ for some random $r_S$.
- $v_R/Q_R$ is a private/public key-pair of a public key cryptosystem allowing data encryption. Hence, data encrypted under $Q_R$ can only be decrypted by providing $v_R$. $Q_R$ has been OAEP-encoded into $(X_R, Y_R) = \text{OAEP}(Q_R, r_R)$ for some random $r_R$.
- Let $Y = Y_S \oplus Y_R$. Then, $\{Y, \text{Sign}_{PK_V}(Y)\}$ is a digest-signature tuple which can be validated under $PK_V$.

*4.3. Valued e-Coin Generation*

A valued e-coin is generated through a procedure in which both the customer and the vendor do participate.

1. The customer pays the vendor the price of an e-coin.
2. The customer generates a random private key $v_S$ and the corresponding public one $Q_S$. The customer also generates a random $r_S$ and computes $(X_S, Y_S) = \text{OAEP}_{PA}(Q_S, r_S)$.
3. The customer generates a random private key $v_R$ and the corresponding public one $Q_R$, and computes $(X_R, Y_R) = \text{OAEP}(Q_R, r_R)$ for some random $r_R$ chosen by the customer.
4. The customer computes $Y = Y_S \oplus Y_R$.
5. The customer requests the vendor to compute a blind signature on $Y$. Let $\text{Sign}_{PK_V}(Y)$ be the resulting signature. Hence, $\{Y, \text{Sign}_{PK_V}(Y)\}$ is a digest-signature tuple.

At the end of this process, the customer is in possession of an e-coin tuple as that shown in Equation (3).

*4.4. No-Valued e-Coin Generation*

A no-valued e-coin is generated by the customer on their own.

1. The customer generates a simulated message-signature tuple under vendor's public key. Let $\{Y, \text{Sign}_{PK_V}(Y)\}$ be the simulated tuple.
2. The customer generates a random private key $v_S$ and the corresponding public one $Q_S$. The customer also generates a random $r_S$ and computes $(X_S, Y_S) = \text{OAEP}_{PA}(Q_S, r_S)$.
3. The customer calculates $Y_R = Y \oplus Y_S$, generates a random $X_R$, and computes $(Q_R, r_R) = \text{OAEP}^{-1}(X_R, Y_R)$. If $Q_R$ is not a valid public key, this step is run again taking a different $X_R$.

At the end of this process, the customer is in possession of a partial e-coin tuple as that shown in Equation (4).

$$\{v_S, Q_S, (X_S, Y_S), \varnothing, Q_R, (X_R, Y_R), Y, \text{Sign}_{PK_V}(Y)\}. \tag{4}$$

The private key $v_R$ corresponding to $Q_R$ is not known, and the corresponding part of the tuple is empty ($\varnothing$).

Note that we need a cryptosystem in which the probability of obtaining a valid public key in a pseudo-random manner is relatively high (step 3). More details are given in Section 5.3.

*4.5. Spending an e-Coin*

A customer wishing to acquire some product $P$ whose price is worth the value of an e-coin asks the vendor to engage in the following procedure:

1. The customer sends $\{(X_S, Y_S), (X_R, Y_R), \text{Sign}_{PK_V}(Y)\}$ to the vendor together with a digital signature $\text{Sign}_{Q_S}(\mathcal{H}(CurrentTime||Y_S||Y_R))$ computed with private key $v_S$ ($\mathcal{H}$ is a hash function).
2. The vendor runs $(Q_S, r_S) = \text{OAEP}_{PA}{}^{-1}(X_S, Y_S)$. If the plaintext-awareness checking is met, they check the digital signature received at the previous step under $Q_S$. In case of failure, the e-coin is rejected.
3. The vendor computes $Y = Y_S \oplus Y_R$ and checks that $\{Y, \text{Sign}_{PK_V}(Y)\}$ is a valid digest-signature tuple under vendor's public key $PK_V$.
4. The vendor checks that no e-coin with the same $Y_S$ component has been spent before. In such a case, the previously stored digital signature, which includes the time it was spent for the first time, is returned as a proof of double spending and the transaction is rejected. Otherwise, all the data received at step 1 is stored by the vendor.
5. The vendor computes $(Q_R, r_R) = \text{OAEP}^{-1}(X_R, Y_R)$.
6. The vendor encrypts the product $P$ under public key $Q_R$ (creating a digital envelope if $P$ is large) and sends the resulting ciphertext to the customer.
7. If the spent e-coin was valued, the customer decrypts the received ciphertext using private key $v_R$, getting $P$ as a result. Otherwise, this step is skipped and the customer does not get any product.

## 5. Cryptosystems Choice

This section provides an assessment on the features to be provided by the cryptosystems chosen to implement the paradigm.

*5.1. Cryptosystem for Vendor'S Key-Pair*

In the described digital cash paradigm, the vendor is in possession of a key-pair whose public key has been denoted as $PK_V$. This key-pair is used for the generation of the $\{Y, \text{Sign}_{PK_V}(Y)\}$ digest-signature tuple. This tuple is generated differently depending on whether the generated e-coin is valued or not. More precisely,

- If the e-coin is valued, the customer computes $Y$ and requires the vendor to compute a blind signature on it (Section 4.3, step 5).

- If the e-coin is no-valued, the tuple is simulated by the customer. The vendor does not take part in this process (Section 4.4, step 1).

Hence, the signature scheme chosen for such $\{Y, \text{Sign}_{PK_V}(Y)\}$ tuples has to enable both:

- The computation of *blind* signatures.
- The generation of *simulated* digest-signature tuples.

Next, we discuss two feasible options.

### 5.1.1. RSA Signatures

Given an RSA [21] private key, $d$, and the corresponding public one $(N, e)$, an RSA digital signature over $M$ is computed from secret key $d$ as $S = M^d \pmod{N}$. The resulting $\{M, S\}$ digest-signature tuple is verified by checking whether $M$ equals $S^e \pmod{N}$.

RSA blind signatures [1] can be issued through the following protocol:

1. Alice chooses a random $R \in \mathbb{Z}_N$ and computes $\overline{M} = M \cdot R^e \pmod{N}$ and sends $\overline{M}$ to Bob (operator $\cdot$ denotes the integer modular multiplication).
2. Bob computes $\overline{S} = \overline{M}^d \pmod{N}$ and sends $\overline{S}$ to Alice.
3. Alice computes $S = \overline{S} \cdot R^{-1} \pmod{N}$ obtaining signature $S$ on $M$.

RSA digest-signature tuples can be simulated by taking a random $S \in \mathbb{Z}_N$ and then computing $M = S^e \pmod{N}$. In a typical signature, $M$ is the hash digest of the piece of data to be signed. Hence, obtaining a message signed by simulation further requires inversion of such hash function which is unfeasible.

We can enable RSA signatures with the construction presented in Section 3.3. By signing the $Y$ component of an OAEP-encoded message, we allow simulated signatures over pseudo-random pieces of data.

### 5.1.2. Boldyreva Signatures

Boldyreva digital signatures [22] are discrete-logarithm based and implemented over a so-called *Gap Diffie–Hellman* (GDH) group. In a GDH group, the *Diffie–Hellman problem* is difficult, namely, given $g^a$ and $g^b$, computing $g^{ab}$ is assumed to be hard. However, the *Decisional Diffie–Hellman problem* is easy to solve, namely, given $g^a, g^b$, and $g^c$, it is easy to decide whether $c = ab$.

A GDH group of large prime order $q$ has to be chosen. Let $g$ be a generator of such group. A private key is generated by choosing a random $x \in \{0, \ldots, q-1\}$. The corresponding public key is computed as $y = g^x$.

A digital signature over a digest $M$ is computed as $S = M^x$.

A digest-signature tuple $\{M, S\}$ is validated under public key $y$ by checking whether $\{M, y, S\}$ is a GDH-tuple, that is, $\log_g M \cdot \log_g y = \log_g S$.

This signature scheme allows the computation of blind signatures throughout the following procedure:

1. Alice chooses $r \in \{0, \ldots, q-1\}$ and computes $\overline{M} = M \cdot g^r$. Then she sends $\overline{M}$ to Bob.
2. Bob computes $\overline{S} = \overline{M}^x$ and sends $\overline{\sigma}$ back to Alice.
3. Finally, Alice computes $S = \overline{S} \cdot y^{-r}$ which is a digital signature over $M$.

Given the group generator $g$ and a public key $y$ $(y = g^x)$, a Boldyreva digest-signature tuple is simulated by taking a random integer $t$ and computing $M = g^t$ and $S = y^t$. Tuple $\{M, S\}$ is a simulated digest-signature tuple. The construction in Section 3.3 allows us to link simulated signatures to pseudo-random pieces of data.

### 5.2. Cryptosystem for e-Coin Transaction Signature

When a customer spends an e-coin (Section 4.5, step 1) they make use of the $v_S$ private key to issue a digital signature that will be validated under $Q_S$. This $\{v_S, Q_S\}$ key-pair is always generated through the traditional (private key first, public key next) procedure. Hence, any cryptosystem allowing digital signature computation can be chosen.

### 5.3. Cryptosystem for Product Encryption

When the customer spends an e-coin to acquire some product, the vendor encrypts it under the $Q_R$ public key included in the spent e-coin (Section 4.5, step 6). The customer is only able to decrypt such ciphertext if they know the corresponding private key. This public key is generated differently depending on whether the e-coin it is contained in is valued or not.

- If the e-coin is valued, the customer generates private key $v_R$ and then the corresponding public one $Q_R$ (Section 4.3, step 3).
- If the e-coin is no-valued, public key $Q_R$ is obtained pseudo-randomly (Section 4.4, step 3).

Therefore, the cryptosystem for such $\{v_R, Q_R\}$ key-pairs must satisfy the following requirements:

- It allows public key data encryption;
- It provides a relatively high probability of obtaining a valid public key by means of a pseudo-random process;
- It cannot be determined whether a given public key has been generated together with its private counterpart (Section 4.3 step 3) or through a pseudo-random process (Section 4.4, step 3).

The RSA [21] cryptosystem would not be a suitable option since the probability that a pseudo-random integer $N$ is composed of two large prime factors is rather low.

We next detail two suitable public key encryption schemes.

#### 5.3.1. ECIES

The Elliptic Curve Integrated Encryption Scheme (ECIES) [23] is an elliptic curve-based public key encryption scheme whose security holds on the assumed intractability of the *Elliptic Curve Discrete Logarithm Problem* (ECDLP).

Such a cryptosystem is set by choosing an elliptic curve $E$ represented as an expression of the form shown in Equation (5):

$$Y^2 = X^3 + AX + B, \tag{5}$$

with $A, B$ being elements of a finite field $\mathbb{F}$ such that its set of points $E(\mathbb{F})$ has a cardinality divisible by a large prime $q$. An order-$q$ point $P$ of $E(\mathbb{F})$ is also chosen. Throughout this section, we assume $q$ is prime.

An ECIES private key is generated by choosing a random $v \in \{0, \ldots, q-1\}$. The corresponding public key is the point of $E(\mathbb{F})$ computed as $Q = vP$.

The probability that a random point $(x, y) \in \mathbb{F} \times \mathbb{F}$ belongs to $E(\mathbb{F})$ is negligible since its components should satisfy Equation (5). Nevertheless, this drawback can be addressed by representing elliptic curve points in compressed form. A point $(x, y) \in E(\mathbb{F})$ can be represented as $(x, b)$ with $b$ being a Boolean indicating whether $y > -y$. In this way, a randomly generated compressed point $(x, b)$ belongs to $E(\mathbb{F})$ (and hence it is a public key), if its $x$-component satisfies that $x^3 + Ax + B$ is a quadratic residue in $\mathbb{F}$. This happens with a close to $1/2$ probability [17].

#### 5.3.2. ElGamal

ElGamal [24] is a public key cryptosystem whose security holds on the assumed intractability of the *Discrete Logarithm Problem* (DLP).

This cryptosystem is set by choosing a large prime $q$ satisfying that $p = 2q + 1$ is also prime. The cryptosystem is built on the order-$q$ multiplicative subgroup of $\mathbb{Z}_p^*$. An order-$q$ element $g$ is chosen during the setup.

A private key is generated by choosing a random $x \in \{0, \ldots, q-1\}$ and the corresponding public key is computed as $y = g^x \pmod{p}$.

A randomly selected element from $\mathbb{Z}_p^*$ turns out to be a public key if its order is $q$. This happens exactly with a 1/2 probability since $\mathbb{Z}_p^*$ contains exactly $p - 1 = 2q$ elements with $q$ of them having the desired order.

## 6. Security Analysis

A digital cash system like the one presented in this paper should satisfy the following security requirements:

1. Valued e-coins cannot be forged by malicious customers;
2. E-coins cannot be double-spent;
3. Customers cannot be falsely accused of double-spending an e-coin.

Although the vendor can be assumed to be a somehow trusted party, Req. 3 is still needed to prevent malicious double-spenders from claiming they are being accused falsely. The following lemmas address the fulfillment of the enumerated requirements.

**Lemma 2.** *Valued e-coins of the proposed digital cash paradigm cannot be forged.*

**Proof.** Let us recall that an e-coin is a tuple of the form shown in Equation (3). An e-coin can only be spent if private key $v_S$ is known. Otherwise, the digital signature required at step 1 of the "Spending" protocol cannot be computed. Hence, $Q_S$ must be generated together with $v_S$, and the $Y_S$ component of its OAEP encoding (with plaintext awareness) is obtained pseudo-randomly by calling $\text{OAEP}_{\text{PA}}(Q_S, r_S)$ for some random $r_S$. Hence, the $Y_S$ component cannot be chosen by a dishonest party aiming to forge an e-coin. Note also that the $Y_S$ component of spent e-coins is checked not to be part of an already spent e-coin. In this way, there is no point in taking the $Y_S$ component of a new e-coin from an existing one.

If the forger then simulates the $\{Y, \text{Sign}_{PK_V}(Y)\}$ digest-signature pair, the resulting $Y$ cannot be chosen (otherwise the underlying signature scheme would be forgeable), so that $Y_R = Y \oplus Y_S$ can neither be chosen and, after taking any $X_R$, the public key $Q_R$ obtained from $(Q_R, r_R) = \text{OAEP}^{-1}(X_R, Y_R)$ is pseudo-random and its private key remains unknown. In this way, the resulting e-coin is no-valued. Lemma 1 guarantees that given $Y_R$ and some chosen $Q_R$, finding a $\{r_R, X_R\}$ pair satisfying the previous expression is unfeasible.

Alternatively, the forger could generate a $v_R/Q_R$ key-pair and OAEP-encode it into $(X_R, Y_R)$. In this case, the obtained $Y_R$ component is pseudo-random and so the resulting $Y = Y_S \oplus Y_R$ is. Hence, the signature $\text{Sign}_{PK_V}(Y)$ over $Y$ cannot be obtained by the forger without the participation of the vendor. □

**Lemma 3.** *E-coins of the proposed digital cash system cannot be double-spent.*

**Proof.** When an e-coin is spent, the vendor stores a record which includes its $Y_S$ component. Hence, any attempt to spend the same e-coin in the future will be detected. □

**Lemma 4.** *An honest customer cannot be falsely accused of being a double-spender by a dishonest vendor.*

**Proof.** Customers spending an e-coin are required to digitally sign a timestamped sequence using the $v_S$ private key. This digital signature can be validated under public key $Q_S$. Only the customer who generated an e-coin knows its $v_S$ secret key.

A vendor claiming that an e-coin is being double-spent is required to provide the signed timestamped sequence of the first time the e-coin was spent (Section 4.5, step 4). If the claim is false, they will be unable to provide it. □

## 7. Experimental Results

The proposed paradigm has been validated through a prototype implemented in Java. Cryptographic operations involving large integers use the `java.math.BigInteger`

library. Hash digests have been computed using the SHA-224 [25] function. Regarding the employed cryptosystems, we have chosen the following:

- *Vendor's key-pair (Section 5.1):* RSA with 2048 bit keys.
- *Cryptosystem for e-coin transaction signature (Section 5.2):* ECDSA [26] with 224 bit keys.
- *Cryptosystem for product encryption (Section 5.3):* ECIES with 224 bit keys.

Our experiments have measured the running time of the "Valued e-coin generation" (Section 4.3), "No-valued e-coin generation" (Section 4.4), and "Spending an e-coin" (Section 4.5) procedures. The prototype has been run on several personal computers. Average running times from 500 executions have been measured. As expected, computers with a faster processor lead to better running times. We have also observed that the running time benefits from parallel execution mode.

Table 1 shows the average running time of the "Valued e-coin generation" and "No-valued e-coin generation" procedures. Let us recall that the generation of valued e-coins involves both the customer and the server (which is required to compute a blind signature) while the procedure for generating no-valued ones is run entirely by the customer. The table shows that the generation of a no-valued e-coin takes some more time than a valued one. This is due to the fact that step 3 of the procedure for generating no-valued e-coins sometimes has to be run more than one time. In our experiments, in which we have implemented the ECIES cryptosystem, there is a 50% chance of having to run it again. In the fastest tested processor, in parallel mode, generation of a valued e-coin and a no-valued one takes around 3 and 4 ms, respectively, leading to generation rates of 333 and 250 e-coins per second, respectively.

**Table 1.** "E-coin generation" running times (in milliseconds).

| Processor | System Server & Client | | | Valued e-Coin | | No-Valued e-Coin | |
| | Cores | Threads | GHz | Serial | Parallel | Serial | Parallel |
|---|---|---|---|---|---|---|---|
| AMD Athlon | 4 | 4 | 2.80 | 49.28 | 13.62 | 68.25 | 20.39 |
| Intel i5-8350U | 4 | 8 | 1.70–3.60 | 21.40 | 5.23 | 32.69 | 7.69 |
| Intel i7-6700 | 4 | 8 | 3.40–4.00 | 20.50 | 4.71 | 28.95 | 7.27 |
| Intel i7-8700 | 6 | 12 | 3.20–4.60 | 18.75 | 4.66 | 28.41 | 6.31 |
| AMD Ryzen 7 | 8 | 16 | 3.70–4.30 | 23.18 | 3.05 | 32.86 | 3.97 |

Table 2 shows the running time of the "Spending an e-coin" procedure at the vendor part. We focus on this part of the process due to the fact that a vendor may receive a lot of concurrent payments. We do not distinguish between spending a valued or a no-valued e-coin since the procedure is exactly the same in both cases. The fastest running time, obtained on an AMD Ryzen 7 processor in parallel mode, indicates that receiving an e-coin payment takes 2.69 ms, so that around 371 payments can be processed in a single second.

**Table 2.** "Spending an e-coin" running times (in milliseconds).

| Processor | System Server | | | Serial | Parallel |
| | Cores | Threads | GHz | | |
|---|---|---|---|---|---|
| AMD Athlon | 4 | 4 | 2.80 | 51.24 | 13.99 |
| Intel i5-8350U | 4 | 8 | 1.70-3.60 | 26.67 | 7.37 |
| Intel i7-6700 | 4 | 8 | 3.40-4.00 | 22.68 | 5.12 |
| Intel i7-8700 | 6 | 12 | 3.20-4.60 | 20.69 | 3.59 |
| AMD Ryzen 7 | 8 | 16 | 3.70-4.30 | 25.41 | 2.69 |

## 8. Conclusions

This paper has presented a novel digital cash paradigm in which customers are able to generate no-valued e-coins by themselves. Such no-valued e-coins can be spent like regular valued ones in such a way that the vendor receiving a payment is unable to distinguish between both situations. The customer only receives the requested digital

product when the spent e-coin is a valued one. This new paradigm fits in scenarios in which customers may wish to mask their consumption patterns through bogus transactions like pay-per-view TV or music platforms. The paradigm has already proven its validity in privacy-preserving pay-by-phone parking systems enabling drivers with the possibility of keeping their expected parking time secret.

In our future research, we plan to investigate the design of privacy-preserving protocols, which include the presented digital cash paradigm as a building block.

**Author Contributions:** Conceptualization, R.B. and F.S.; methodology, R.B. and F.S.; validation, R.B. and F.S.; formal analysis, F.S.; writing—original draft preparation, F.S.; writing—review and editing, R.B.; funding acquisition, F.S. All authors have read and agreed to the published version of the manuscript.

**Funding:** This research was funded by the Spanish Ministry of Science, Innovation and Universities grant number MTM2017-83271-R.

**Institutional Review Board Statement:** Not applicable.

**Informed Consent Statement:** Not applicable.

**Data Availability Statement:** Data is contained within the article.

**Conflicts of Interest:** The authors declare no conflict of interest. The funders had no role in the design of the study; in the collection, analyses, or interpretation of data; in the writing of the manuscript, or in the decision to publish the results.

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
