# Peer review of "A Digital Cash Paradigm with Valued and No-Valued e-Coins"

_applsci, doi:10.3390/app11219892_

Round 1

Reviewer 1 Report

This paper presents a new digital cash paradigm that includes the so-called no-valued e-coins, which are e-coins that can be generated free of charge by customers.
The overall organization is clear and concise.

Suggestions: 

1. line 44: I think this statement "We refer the reader to [8] where a broad review of electronic cash systems can be found." is not suitable. It should be deleted.
2. This paper provides a formal analysis of its security. Not only security could be analyzed. Others, for example, performance, should also compare with other approaches. 
3. References 7 should be : "Wood, G. Ethereum: A secure decentralised generalised transaction ledger. Ethereum project yellow paper, 2014, 151.2014: 1-32."  The original one lack of some information.
4. Reference 16 should be: "ELGAMAL, T. A public key cryptosystem and a signature scheme based on discrete logarithms. IEEE transactions on information theory, 1985, 31.4: 469-472."  The original one lack of year.

Reviewer 2 Report

Introduction - The authors  comprehensively introduce to the readers a general concept of e-coins.

The main issue with this manuscript that is the literature review section is missing. All references (except one) are more than 10-15 years old. The literature review should be done to show what research studies exist in this area and how researchers tries to solve the problem of efficient e-coin generation and transaction.

Line 46 – no need in subsection. Just continue the text.

Line 55 – can you collaborate more about initial design

Line 58-60 – this information is given 2 paragraphs above. Replace with “in this paper we propose how the key size limitation can be avoided/eliminated”

Line 65 - OAEP-based – abbr needs definition.

Line 196 – the equation should be numbered (1, 2 and so on). The indexes S and R should be defined. By whom generated the public/private key pair for S and R components?

Line 198 – say that is generated by customer/requester (same for 202)

Line 203,215,224 – “they generate…” who generates r with index S and r with index R? If random numbers generated both by customer and server side how we can ensure that they are same/not the same?

Comment: public key pair generated only on user side, how we can ensure trust?

The steps involved in the process are described on quite high level

Line 278 - M = M Re (mod N). What math operator is used between M and R? Is it xoring?

Interesting approach but needs  an experimental validation since the private key V with index R that omitted in the proposed method can cause computational difficulties.

Can be accepted after including literature review section.

Round 2

Reviewer 1 Report

none